# Effect of government's environmental attention on corporate philanthropy based on the institutional theory: Evidence from China's heavily polluting companies

Dongli Cao[1]*, Chunxian Nie[2]

1 School of Law, Southwestern University of Finance and Economics, Chengdu, Sichuan Province, China,
2 Institute of Western China Economic Research, Southwestern University of Finance and Economics, Chengdu, Sichuan Province, China

* 1200201z1001@smail.swufe.edu.cn

**Data Availability Statement:** All relevant data are within the Supporting information files and also available from the GITHUB database (DOI:https://doi.org/10.5281/zenodo.13338237).

## Abstract

The underlying motivation behind corporate philanthropy (CP) is subject to multiple interpretations. For emerging markets, traditional interpretations based on a perspective of interest exchange often fall short. The institutional environment in China is characterized by uncertainty, particularly in the field of environmental protection, where the government's attention, a scarce institutional resource, can influence the behaviors of heavily polluting companies. The government's environmental attention (GEA) may be subject to campaign-style investment, which provides an opportunity for research. This study attempts to use institutional theory to construct a difference-in-differences model using the Central Inspection of Environmental Protection as an exogenous shock. It examines the causal impact of GEA on CP, observing that GEA has a positive effect on CP, especially for large companies, companies in less competitive industries, companies located in high-pollution areas, and companies with high environmental expenditure. The results indicate that campaign-style institutional changes provide a non-transactional political motivation for CP. In addition, we discuss the mechanism, which is the expectation change. Campaign-style institutional changes work by altering corporations' expectations of the future institutional environment. We verify this mechanism from two perspectives, namely, the intensity of changes caused by uncertainty and corporate sensitivity to changes. Our results remain robust when subject to several tests based on different identification hypotheses and alternative measures. Unlike the existing literature on the political connections driving CP in China, this study, based on the institutional theory framework, reveals that campaign-style institutional changes constitute a non-transactional, unidirectional political motivation, which is a significant driving factor for CP in a Chinese context. This study evaluates the importance of GEA and reveals the non-transactional political motivations from the institutional theory perspective. This not only enriches existing discussions on the political motivations of CP but also broadens our understanding of the relationship between institutional uncertainty and CP in emerging markets.

**Funding:** The author(s) received no specific funding for this work.

**Competing interests:** The authors have declared that no competing interests exist.

## 1. Introduction

Corporate philanthropy (CP) has garnered global recognition in recent years. According to the report of The Giving USA Foundation in 2018, CP in the United States exceeded $20 billion in 2017, representing an approximately 8% increase over the previous year. The Global Philanthropy Tracker reported that the total charitable outflow from 47 countries reached $70 billion in 2020. Over 200 large companies worldwide are engaged in CP, with US companies reaching $471.44 billion toward CP. Before 2020, the annual growth rate of CP consistently remained above 5%. Even in 2020, despite the COVID-19 pandemic, total contributions fell by only 0.5%, demonstrating the resilience of the philanthropic sector in the face of crises. The data also indicate that CP is transitioning from temporary donations to more strategic and targeted philanthropy, focusing on a few signature issues that align with the parent company's values, expertise, and business goals. Furthermore, driven by global fluctuations in energy costs and supply shortages, CP incorporates climate concerns.

CP not only provides advertising effects and tax benefits [1, 2], increasing companies' economic advantages, but emerging market companies also prefer it because CP can generally be deducted before income tax [3] and is more direct and transparent [4]. The 2023 China Philanthropy List included 1363 companies, with a total donation amount of $2.778 billion. According to the 2023 China Corporate Foundation Observation Report, Chinese corporate foundations are growing steadily. As of the end of 2022, there were 1850 Chinese corporate foundations, mainly located in Guangdong, Beijing, Zhejiang, Shanghai, and Jiangsu, with the most significant investments going toward education, rural revitalization, and public services. Evidently, Chinese companies' participation and contribution levels in the philanthropic field are continually increasing.

The motivational factors driving CP have long been a topic of interest among scholars. Previous research on the motivations generally provides two explanations: altruistic and instrumental. The former posits that CP is primarily rooted in altruism, where companies seek to help others and benefit society with no expectation of a return [5–7], thus embodying the values of "corporate citizens" [8]. However, this perspective has gradually been abandoned over time, as CP seldom stems purely from altruism in the 21st century. It is highly uncommon for CP not to expect any return. CP is often a combination of strategic considerations with a touch of altruism. Choi and Wang [9] observed that senior executives with charitable and integrity values are more inclined to give back to society via CP. Cha and Rew [10] highlighted that a CEO's personal values often influence their philanthropic decisions. Lin et al. [11] found that companies with more diverse boards are likelier to engage in charitable endeavors. Nevertheless, these studies do not acknowledge pure altruism but discuss that CP may still be influenced by altruism—even when it may no longer be the primary driving force. Given that CP could prioritize managers' self-interest over that of shareholders [12] and could deplete cash flow, potentially hindering innovative investments [13], it is now widely accepted that the most important motivation is to fulfill the company's strategic needs, that is, the instrumental explanation.

The instrumental explanation, which is the currently prevailing perspective, posits that CP is not driven by altruism. Instead, it is a necessary tool for attaining specific objectives [14]. Hemingway and Maclagan [15] proposed that the instrumental motivations for CP might include several aspects, primarily economic, political, and managerial, and these motivations might manifest at multiple levels [16]. The economic motive suggests that CP is a company's strategic deployment in response to the competitive environment, which aims to build a good reputation [3, 17], enhance the competitive environment [18, 19], and maximize corporate profits [20]. In the long run, CP can create a feedback effect [21, 22], effectively enhancing

corporate economic performance [23, 24], lowering the actual tax burden [25, 26], and improving the financial condition [27]. CP can also exhibit good profitability, providing companies with an advantage in securing external financing [18]. Additionally, CP can help mitigate the negative reputation of polluting companies by addressing violations in environmental and health safety regulations [17].

Political motive refers to companies using CP as a strategy to gain political advantages, such as providing financial support for political candidates' campaigns, assisting local governments in achieving key policies [28], and cultivating strong political connections to obtain favorable policy or regulatory outcomes [29, 30]. Bertrand et al. [31] highlighted that without disclosure requirements, CP might serve as a means of political influence, financed by taxpayers and undiscovered by voters. Political reciprocity may also facilitate business investment [32] and government project opportunities [33]. Lin et al. [34] observed that Chinese companies tend to use CP to establish political networks, thereby assisting companies in securing local government financial subsidies.

The managerial motive suggests that CP represents active social responsibility by companies. This can foster employees' trust in the organization, decrease formal control costs [35], and, therefore, bolster employee commitment, increasing their willingness to actively work and contribute to the company [36, 37].

The behavior patterns shown by companies in emerging markets significantly differ from those in developed countries, particularly regarding CP [38]. This is because of the notable differences in corporate governance, organizational structure, and social background. China, as the world's largest emerging market, offers valuable research material to elucidate the underlying motivations for CP in emerging markets. It is widely accepted that while traditional Chinese culture emphasizes "helping the world when successful" and "supporting each other in times of difficulty," and contemporary socialism with Chinese characteristics also promotes social harmony and the selfless "Lei Feng spirit" [39], CP is hardly ever driven by pure altruism. Companies often use media reports to present a positive image of fulfilling social responsibilities [40], amplifying the visibility of CP, attracting and obtaining key resources from stakeholders, and boosting corporate performance [41]. Therefore, the altruistic motive does not hold true in China.

Prior research on instrumental motives has predominantly focused on economic and managerial motives. However, economic motive may not be the main impetus for CP in China [42]. This is because many studies overlook one critical fact: the Chinese government has the most significant influence on economic development and corporate management. Thus, political motive might be the most crucial driver for CP. The Chinese government exercises control over the allocation of core resources and has an irreplaceable influence on corporate operations [43, 44]. As a strategic tool, CP exhibits different forms under different national, institutional, and regulatory constraints. Given the uniqueness of the Chinese government, which assumes multiple roles, including referee, stakeholder, and audience [45], and the absence of transparency in government decision-making, along with the government's absolute dominance in economic affairs, corporate development must align with the government's preferences and address the issues that currently hold the government's highest level of concern [46].

Previous studies on political motives almost entirely focus on political connections. Yang and Tang [47] analyzed the philanthropy of Chinese private entrepreneurs utilizing data from a national survey in 2012. The survey was jointly conducted by the All-China Federation of Industry and Commerce, the State Administration of Industry and Commerce, the United Front Work Department of the Central Committee of the Communist Party of China, and the China Society of Private Economy at the Chinese Academy of Social Sciences. The study

revealed that Chinese private entrepreneurs with formal connections to government agencies are more inclined to contribute toward CP. However, this study defined political connections as whether entrepreneurs have a formal political status, which may lead to potential reverse causality identification. Furthermore, because the survey did not clarify whether the entrepreneurs' donations were public or confidential, it is challenging to distinguish whether the motivation is altruistic or instrumental. This ambiguity severely undermines the reliability of the conclusions on political motivations. Dang et al. [48] went further by categorizing political connections into symbolic and material types, overcoming the potential endogeneity problem brought about by only considering formal political statuses, such as deputies to the People's Congress and members of the Chinese People's Political Consultative Conference. Nevertheless, their empirical data comprised listed companies from 2008 to 2015, which is relatively outdated and mainly focused on the period before 2013. The lack of coverage for the period after the 18th National Congress of the Communist Party of China makes it challenging to determine whether political connections still hold after 2013.

Jia and Zhang [49] contend that China's distinctive system has fostered a political market where political resources and corporate funds are exchanged. From this political market standpoint, they examined CP and observed that local officials' political turnover creates opportunities for companies to capture the attention of newly appointed officials through CP. Similarly, Xu and Liu [50] observed the political motive from a resource exchange perspective. They utilized data from Chinese A-share listed companies from 2008 to 2013 and found that competition for political resources is a critical factor driving CP. Nevertheless, these studies primarily rely on the framework of interest exchange and predominately analyze data from before 2013, thereby disregarding the anti-corruption campaign following the 18th NCCPC. Consequently, these studies fail to provide explanatory power for the challenges associated with political connections after 2013. Hao et al. [51] found that China's anti-corruption campaign in 2013 significantly reduced CP from listed companies with strong political connections. Additionally, the government took measures to decrease fiscal subsidies to these companies. This suggests that the explanatory power of political connections is rapidly declining. However, Hao et al.'s study could not answer whether other political motives outside of political connections still drove CP after the anti-corruption campaign. Therefore, we chose to begin our analysis from the standpoint of institutional theory and examine the one-way political motive of government's environmental attention (GEA), extending the political motive beyond the two-way political connections.

In addition to interest exchange, several studies focus on the turnover of local officials. Dai et al. [52] examined the turnover of local officials and found that it significantly increases CP. Companies acquire multiple benefits, such as credit, subsidies, and investment opportunities, indicating that CP is a disguised form of "political donation." Liu et al. [53] also used the turnover of local city leaders as their research material. They contend that companies in emerging markets are often forced to make decisions under heightened political uncertainty. The turnover of local officials is an institutional change, resulting in uncertainty. Additionally, they examined the impact of institutional changes on CP. Alternatively, this study considers the turnover of local officials as the primary source of instability in Chinese institutions, which also exhibits deficiencies. The regulation of rotating local officials every three to five years has been implemented for several decades in China. Chinese companies have adopted this normative regulation as a standard practice. In other words, the turnover of local officials may not currently bring significant uncertainty to companies.

Existing research has the following shortcomings: (1) The research on examining the motives for CP in the Chinese context tends to excessively fixate on economic or managerial motives [17, 18, 54, 55] while neglecting political motives. There is a lack of consideration for

government factors, and the role of institutional changes has not been adequately analyzed in the literature; (2) A few scholars who argue that CP is based on political motives in China only focus on political connections [45, 47, 49, 50] or analyze heterogeneity from the perspective of state-owned enterprises (SOEs) and private enterprises [56, 57]. Political connections between companies and local governments exist only partially [58] and are a type of two-way interaction. Political connections serve as a means for local governments to intervene in companies and a crucial channel for companies to obtain government resources [59]. However, after the severe crackdown on corruption after the 18th NCCPC, the boundary between political connections and corruption is unclear, and two-way interactions have decreased. The existing literature is deficient in research concerning one-way, non-transactional political motives; (3) Studies exploring political motives from the perspective of uncertainty primarily consider the turnover of local officials as the source of uncertainty [52, 53], neglecting the fact that this institutional change has become normalized. There is little research examining the political motives from the standpoint of a more unpredictable, campaign-style governance.

This study uses the Central Inspection of Environmental Protection (CIEP), an exogenous event, as a quasi-natural experiment to measure the campaign-style institutional change in GEA. In recent decades, the rapid economic development in several developing countries has led to the emergence of environmental pollution. Environmental pollution has a significant impact on residents' health [60], infrastructure construction [61], urbanization process [62], and company governance [63], among others. The Chinese government has initiated efforts to address the relationship between economic development and environmental protection, thus fostering the promotion of green economic transition. Nevertheless, government resources and attention are constrained, with government attention scarcer than external information. The allocation of attention significantly impacts actual decision-making [64]. The more concentrated the government's attention is on a specific area, the more resources and policies it will allocate to that area. The government's attention also represents the government's most urgent needs and preferences. Since 2004, the Chinese government has been promoting the concept of a "harmonious society" and encouraging companies to take on social responsibilities [65]. In the name of public interest, CP is an inevitable choice for companies to attract the government's attention for long-term development. To secure enough resources or to avoid certain regulations or penalties, companies ensure their legitimacy through CP. The greatest institutional uncertainty often does not arise from the turnover of local officials [66] but from the campaign-style governance brought about by the short-term tilt of the government's attention.

Nevertheless, there is a challenge in identifying causality directly between GEA and CP. First, GEA and CP may be driven by unobservable common characteristics, resulting in an omitted variable bias. Second, performance changes in corporate social responsibility may lead to changes in GEA in a local area, which is a typical reverse causality. Therefore, the results obtained from the standard ordinary least squares estimation may not be able to reveal the causal relationship. To overcome this problem, we used CIEP as a quasi-natural experiment to construct a difference-in-differences (DID) model. The treatment group comprises companies in heavily polluting industries since 2016, whereas the control group consists of companies that do not satisfy this condition. Prior to the implementation of CIEP, there was no significant increase in CP. After 2016, GEA significantly enhances CP. These estimates are reliable for controlling various confounding factors, such as firm age, equity nature, firm size, company governance, and financial conditions, as well as regional, industry, and year fixed effects. This suggests that GEA, as a campaign-style institutional change, indeed serves as a political motive for CP. Furthermore, the application of the DID model is contingent upon a series of basic model assumptions. We conducted several tests, including parallel trends test, placebo test,

propensity score matching (PSM)-DID model, controlling for concurrent parallel policies, and other robustness tests, to verify these assumptions. Our results showed robustness. Subsequently, we conducted a further discussion. We believe that the mechanism by which GEA enhances CP is that GEA changes corporate expectations, prompting companies to actively seek legitimacy and cope with institutional uncertainty. We tested the mechanism with changes in intensity and corporate sensitivity by introducing details of CIEP manually collected. In addition, we conducted heterogeneity analysis and found that the enhancing effect of GEA on CP is particularly prominent for large companies, companies in less competitive industries, companies located in high-pollution areas, and companies with high environmental expenditure.

This study makes three primary contributions. First, it expands the existing literature on CP, particularly regarding the political motives for CP. Prior research on CP in China tends to oversimplify political motives as solely political connections, disregarding the political driving force that government factors can generate apart from such connections. The role of non-transactional political motives has not been well validated. Fan et al. [67] observed that political connections can help companies more easily acquire bank loans, Wu et al. [59] determined that political connections can bring tax benefits, and Xu et al. [68] found that political connections can even act as substitutes when formal institutions are imperfect. Moreover, they can safeguard property rights when companies encounter challenges and mitigate unfair treatment. As mentioned earlier, several studies contend that the essence of political connections is a form of interest exchange [45, 47–50, 54], offering political incentives for CP. All these studies highlight that political connections are two-way interactions. Companies acquire scarce resources provided by the government while simultaneously fulfilling the exchange requirements set by local officials. However, since the 18th NCCPC launched a strong anti-corruption campaign, the explanatory power of the interest exchange theory presents weaknesses. We have, therefore, abandoned the notion of interest exchange and instead focused on the non-transactional, unidirectional political motives that had been overlooked in the past. Based on the institutional theory, we introduced GEA to analyze the non-transactional political motives. The empirical results demonstrate that GEA is a key influencing factor driving CP, hence addressing the research gap in unidirectional political motives. Additionally, we thoroughly investigated its mechanism, that is, expectation change, expanding on the current literature.

Second, we expanded the discourse around institutional uncertainty. Prior research usually assumes a relatively stable institutional environment for CP. There is a paucity of literature on institutional changes, especially temporary, unpredictable, non-normal institutional changes. Unlike Liu et al. [53], we examined the impact of campaign-style institutional changes on CP. Considering China's distinct institutional background, conventional government behavior is easy for companies to predict and incorporate into their established development strategies. The turnover of local officials has been a normalized institutional arrangement. Although there are certain random factors causing official mobility, the uncertainty brought about by the overall high mobility of local leaders has significantly decreased, which is the focus of Liu's research. We introduced a newer and more nuanced perspective to CP: campaign-style institutional changes. When conventional governance is ineffective, the Chinese government prefers to use campaign-style governance for remediation. Past practices demonstrate that this is an effective error correction mechanism, but it comes at the expense of increased uncertainty. We posit this campaign-style institutional change leads to higher uncertainty, which will prompt companies to make more charitable donations to ensure that the legality is sufficient to embed in the prevailing institutional environment. Hence, our study offers a novel perspective for the examination of institutional uncertainty and CP.

Third, we enriched the literature regarding the impact of government factors. Our analysis differs from that of Jia and Zhang [49] in this respect. First, we used data from 2003 to 2021, with a particular focus on the years after 2013. We fully believe that the anti-corruption campaign after the 18th NCCPC has had a huge impact on China. Government factors before and after 2013 face completely different institutional environments. Utilizing data only before 2013 will inevitably lead to inaccurate results. Second, after the 18th NCCPC, transactional government factors such as political connections or political-business transactions are difficult to distinguish from corruption. To protect their political lives or avert suspicion, local officials have become increasingly cautious in their attitudes toward building connections with companies. Overemphasizing transactional connections based on interest exchange may not constitute all government factors. Our study utilizes data with a broader and more comprehensive time coverage, thus enriching the discourse on the impact of government factors after the large-scale anti-corruption campaign in 2013.

This study's results provide the government with a novel perspective on understanding CP. Our hope is that the government can guide companies to form a long-term expectation of environmental protection by optimizing governance strategies and facilitating a positive interaction between environmental protection and corporate development. The remainder of this paper is organized as follows: Section 2 provides the institutional background and hypothesis development, Section 3 introduces the materials, model, and variable settings, Section 4 presents the empirical results and hypothesis testing, and Section 5 discusses the mechanism and heterogeneity. Finally, Section 6 concludes the study and provides policy recommendations.

## 2. Institutional background, and hypothesis development

### 2.1 Institutional background

In China, local governments possess substantial discretionary authority over the allocation of critical resources, including land, mineral resources, credit, and import and export quotas, as well as the issuance of various administrative approvals and regulations. Due to the central government's "one-vote-veto" evaluation system, local governments have few incentives to exceed the minimum environmental protection standards [52, 69]. Prior to 2015, China's environmental policies were predominantly focused on "supervising enterprises", with the central government responsible for policy formulation and local governments charged with implementation. However, local governments often fell short in their execution of central environmental policies due to competing governance objectives and distorted incentives within the multi-level administrative hierarchy, leading to implementation issues, such as incomplete enforcement [70], flexible interpretation [71], and free-riding [72].

To counteract the speculative actions of local officials, in July 2015, the 14th meeting of the Central Leading Group for Comprehensively Deepening Reforms approved the "Environmental Protection Inspection Scheme". This initiative, which included the CIEP, shifted the focus of environmental oversight from "supervising enterprises" to "supervising local governments and enterprises". The CIEP was structured around inspection teams led by provincial- or ministerial-level official, with deputy team leader appointed from the ranks of deputy minister of the MEEC. Team members were selected from senior central government departments to ensure the inspection's authority and enforcement.

The CIEP operates through three stages: entry, immersion, and analysis. Initially, the inspection team spends a month in each province conducting inspections and investigations into local environmental issues and the performance of local environmental departments. Upon completing this phase, the team submits an inspection report to the provincial government, which is then required to submit a rectification plan to the State Council within 30 days.

**Table 1. Schedule of CIEP.**

| Batch | Period | Provinces involved | Number of officials interviewed | Total amount of fines (million yuan) |
|---|---|---|---|---|
| Pilot | January-February 2016 | Hebei | 65 | 30 |
| First Batch | July-August 2016 | Heilongjiang, Inner Mongolia, Henan, Jiangsu, Jiangxi, Guangxi, Yunnan, Ningxia | 2241 | 198 |
| Second Batch | November-December 2016 | Beijing, Shanghai, Guangdong, Chongqing, Hubei, Shaanxi, Gansu | 4066 | 243 |
| Third Batch | April-May 2017 | Tianjin, Liaoning, Shanxi, Anhui, Hunan, Fujian, Guizhou | 6079 | 336 |
| Fourth Batch | August-September 2017 | Jilin, Shandong, Zhejiang, Sichuan, Hainan, Qinghai, Tibet, Xinjiang | 4210 | 466 |

The CIEP employs two administrative measures, interrogations and accountability, and two legal measures, environmental fines and detention, to deter local governments and punish polluting enterprises. Additionally, it leverages public participation and media oversight to create a mobilization effect. These three mechanisms work in concert to facilitate the smooth execution of the CIEP.

From January 2016 to September 2017, the CIEP conducted its first round of inspections across all provinces in China in four batches (and the first pilot), as detailed in Table 1. A "look-back" inspection was then carried out in 20 provinces from May to December 2018. During these initial and follow-up inspections, the CIEP penalized over 40,000 companies, imposing fines totaling 2.46 billion yuan, and initiated 2303 investigations. More than 24,000 local officials were held accountable, and over 150,000 ecological issues were directly addressed, leading to improvements in air quality [73] and significant reductions in emissions of PM10, SO2, CO, and NO2 [74].

## 2.2 Hypothesis development

Institutional theory postulates that organizations are passive actors operating within the framework established by institutions [75] and, thus, must adhere to the regulations set by specific institutions [76]. As the largest emerging market in the world, China stands apart from developed countries in that the government directly interferes with the economy, plays a vital role in economic development, and even directly participates in economic activities. The Chinese government has the authority to allocate resources and administrative management, exerting significant influence on corporate behavior. The relationship between the government and firms is characterized by a higher level of complexity and subtlety compared to other countries. Therefore, companies must always prioritize issues that the government highly values to effectively cope with the uncertainty of institutional changes.

China's shift toward high-quality and sustainable development has led to a year-by-year improvement in the government's focus on environmental governance and the green economy. China's central government has placed significant emphasis on environmental protection and governance transformation. Baumgartner and Jones [77] introduced the concept of attention into public policy and government decision-making, posting that attention toward policy decisions is limited, whereas external information is unlimited. Hence, the governance effect in a certain area is often contingent upon how much attention the government has invested. The more attention the government pays to a certain area, the more resources it will allocate to that area. To enhance the implementation of green economic transformation, the Chinese government has invested GEA in this area, making environmental protection a top priority [78]. This ensures the effect of environmental governance; however, in practice, GEA

often adopts the form of campaign-style governance, such as the CIEP examined in this study, which also introduces significant institutional uncertainty.

Given the potential adverse impact of institutional uncertainty on companies [79], such as increased financing constraints [80], amplified stock volatility [81], diminished asset prices [82], and decreased profits [83], companies must take proactive measures. These measures may include decreasing investment levels [84], minimizing cash holdings to mitigate political extraction [85], and employing strategies to obscure information disclosure [86]. However, these measures cannot guarantee legitimacy, and companies often require more targeted manipulation strategies to strengthen their legitimacy. The concept of legitimacy originates from the institutional theory. Suchman [87] defines legitimacy as "the appropriateness and acceptance of an organizational entity's behavior within the system of norms, values, and beliefs constructed by society." Legitimacy is achieved when an organization establishes consistency with the values or behavioral norms approved by society [88]. Particularly in emerging markets like China, the government plays a dominant role in conferring legitimacy [89]. The appropriateness of corporate behavior and whether it meets social expectations depends on government confirmation. According to this theory, legitimacy can be manipulated to a certain degree. Organizations can consolidate their legitimacy through CP [90]. This is one of the most important implications of CP as a non-productive expenditure in China's institutional environment, and it is also why this study specifically examines CP rather than other corporate investments, such as corporate environmental investments. Corporate environmental investments or innovative investments are productive investments that can improve corporate production efficiency or future production capabilities. However, they cannot fully highlight the efforts undertaken by companies to respond to institutional changes initiated by the government. In other words, if this study uses corporate environmental investment as the core variable to be analyzed, it will result in the mixing of productive motives and non-productive motives to manipulate legitimacy, severely interfering with causal identification.

Notably, in China, companies acquiring legitimacy via CP also align with the social background and government needs. After rapid economic development, China has progressively shifted its focus toward social stability and sustainable development. Given the ongoing pursuit of personal promotion by local officials, large amounts of resources have been allocated to economic development, while local governments' funds for social services remain consistently insufficient [91]. CP greatly alleviates the budget constraints of local governments and helps fulfill the political task of developing social services for local governments [45].

Whether it is to receive preferential treatment from the government, such as access to specific permitted industries, bank credit, and subsidies, or to minimize unwarranted intervention, companies must implement strategies to meet the government's preferences and expectations. Particularly in the current context where the Chinese government prioritizes environmental governance, companies choose to present themselves as willing to take on social responsibilities to guarantee their legitimacy. Therefore, we propose the following hypothesis:

**Hypothesis(H1):** Government's environmental attention pronotes corporate philanthropy.

Companies are willing to increase CP to ensure legitimacy and respond to institutional uncertainty. Given the central government's firm stance on environmental governance, a company's willingness to increase charitable donations may directly depend on the intensity of the uncertainty. Therefore, the key to the impact of GEA on CP lies in the severity and duration of the institutional changes. If the central government carries out strong rectification through GEA and normalizes and sustains environmental attention, companies' expectations will be

that the institutional environment will be more uncertain. Furthermore, companies are likely to strengthen their willingness to conduct CP. We propose the following hypothesis:

**Hypothesis(H2)**: Higher and more sustained government's environmental attention further enhances corporate philanthropy.

At the same time, although companies can manipulate legitimacy, the manipulation behavior is also contingent upon the company's sensitivity. The endowment effect theory of behavioral economics suggests that people assign a higher value to what they own, resulting in a heightened sense of pain when they lose it [92]. Companies with better legitimacy endowment may be more sensitive to uncertainty and, thus, are inclined to make more efforts to avoid the adverse consequences of insufficient legitimacy. In general, non-state-owned enterprises (NSOEs) aim for profit maximization. Unlike purely profit-oriented NSOEs, the SOEs are directly managed by the State-Owned Assets Supervision and Administration Commission. They align with the value pursuits of the government and often face more administrative requirements. Therefore, compared with NSOEs, SOEs generally have better legitimacy and exhibit more sensitivity to uncertainty. Furthermore, with the popularity of environmental, social, and governance (ESG) practices in China, ESG metrics can largely reflect the performance status of a company's comprehensive social responsibility. High-ESG-scored companies usually have better governance and help the government fulfill its social responsibilities. Sunken costs and the inertia of legitimacy needs may require high-ESG-scored companies to continue maintaining government trust. This may also make high-ESG-scored companies more sensitive when facing uncertainty. Therefore, we propose the following hypothesis:

**Hypothesis(H3)**: State-owned ownership structure and high ESG scores further enhance the positive role of government environmental attention on corporate philanthropy.

CP necessitates a portion of the company's funds; hence, it must consider practical limitations, namely financial constraints. Typically, unlike small and medium-sized companies or companies in highly competitive industries, large companies and those in low-competitive industries often possess better financial conditions and weaker liquidity constraints. They have sufficient capital to increase charitable donations. Therefore, we propose the following hypothesis:

**Hypothesis(H4):** For large companies and companies in low-competitive industries, the government's environmental attention has a more significant impact on corporate philanthropy.

Alternatively, heterogeneity among companies is also reflected in corporate governance pressures. Outdated production models and high environmental costs imply immense pressure on corporate environmental governance, rendering these companies less adept at responding to uncertainty. Such companies often have a greater need for legitimacy. Therefore, we propose the following hypothesis:

**Hypothesis(H5)**: For companies with greater environmental governance pressure, the government's environmental attention has a more significant impact on corporate philanthropy.

The framework of all the hypotheses in this research is shown in Fig 1.

## 3. Method

### 3.1 Material

Our study focuses on China for two reasons. First, in emerging markets, CP has always been a critical strategy for companies to gain legitimacy, particularly in China [93], where the

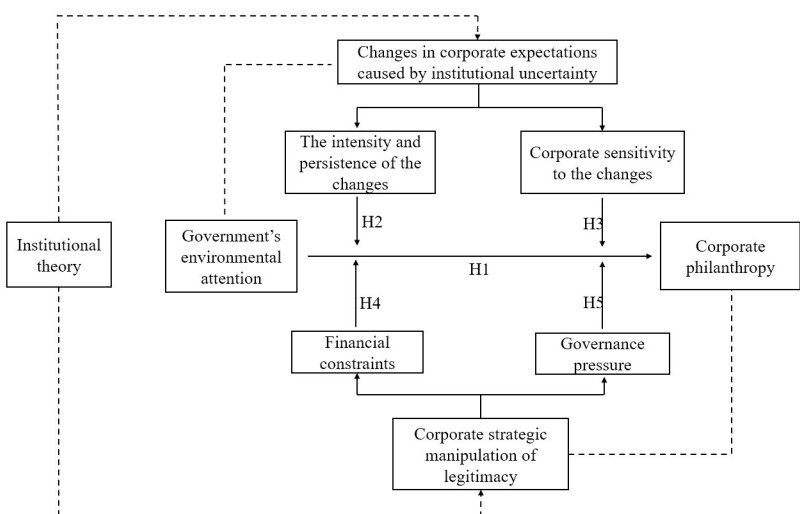

**Fig 1. Framework of research.**

government is a key initiator, stakeholder, and recipient of CP. Simultaneously, CP serves as a vital external resource for the survival and development of Chinese companies. Moreover, information advantages associated with close ties with the government can also provide economic returns, making China a unique and effective research environment. Second, the Chinese central government is resolute in its attitude toward environmental protection and sustainable development. Environmental governance has become a key assessment indicator for the central government's promotion of local officials [94], corporate performance [84], and recognition of high-tech enterprises [95]. The CIEP is implemented by the central government to concentrate GEA, reflecting the temporary and campaign-style institutional changes. The CIEP conducts random inspections without prior notice, preventing companies from predicting the specific time and content of the inspection. This ensures the exogeneity of the quasi-natural experiment. Furthermore, the CIEP inspects not only companies but also local officials, eliminating the potential for CP to be used for the purpose of establishing political connections and preventing interference in causal identification.

We compiled a comprehensive dataset that includes 2126 listed companies in the Chinese A-share market from 2003 to 2021. The data sources are as follows: (1)CP data were derived from the "Non-Business Expenditure, Public Welfare Donation" column disclosed in the financial statement notes within the China Stock Market & Accounting Research (CSMAR) Database; (2) Basic information and financial data of listed companies were sourced from various sub-libraries of the CSMAR; (3) Pollution control expenses were obtained from the environmental governance and green economy sub-libraries of the Chinese Research Data Services Platform; (4) To further examine the mechanism, we manually collected the implementation details of the CIEP from the official website of the Ministry of Ecology and Environment of China, which constitutes a unique data source of this study; (5) ESG scores were obtained from the China Securities ESG rating data in the Wind database; (6) SO2 pollution emission data were sourced from the China Statistical Yearbook and China Environmental Statistics Yearbook. To avoid the bias in the estimation results caused by extreme values in the data, we winsorized all continuous variables with extreme values at the 1% and 99% quantiles.

The initial sample was screened according to the following criteria: (1) excluding financial and insurance listed companies, as they have significantly different financial structures from

other industries; (2) excluding samples that are labeled ST or ST* in this year, as these listed companies, due to continuous financial losses, have significant differences in their philanthropy strategies compared to ordinary companies; (3) excluding samples with missing core data. After screening, this study obtained 40,406 observations.

## 3.2 Model

The core issue of this study is to determine whether GEA increases CP. The CIEP was implemented in January 2016, first piloting a surprise inspection in Hebei Province, followed by rolling it out nationwide. As the CIEP did not announce the location or timing of the inspections in advance, local governments considered the policy unpredictable. The policy may be regarded as a deterrent to all local areas across the country. Therefore, this study employs the CIEP as a quasi-natural experiment representing GEA and constructs a DID model. The first difference comes from the industry level. We posit that companies in heavily polluting industries are inevitably impacted by this policy, whereas those in non-heavily polluting industries serve as the control group. The second difference comes from whether it corresponds to the years 2016 and after. The DID method in this study compares the difference in CP between companies in heavily polluting industries and non-heavily polluting industries before and after the implementation of the CIEP. The model is constructed as follows:

$$CP_{ijct} = \alpha + \beta_1 GEA_{ijct} + X'_{ijct}\varphi + \delta_j + \omega_c + \mu_t + \varepsilon_{ijct} \tag{1}$$

In this model, i represents the company, j represents the industry, c represents the city, and t represents the year.

A core issue of our identification strategy is determining whether there is comparability between heavily polluting and non-heavily polluting industries. Although we have controlled for a large number of corporate characteristic variables and fixed effects, there may still be unknown or unobservable characteristics that render the treatment and control groups incomparable. Therefore, we used PSM-DID and found that the results of the baseline regression still hold. Given the existence of some environmental governance policies before and after 2016, which might result in estimation bias, we controlled for these policies and found that they had no impact on our conclusions. We also included company fixed effects to avoid omitted corporate characteristics and controlled for interaction fixed effects to prevent time-varying factors at the regional or industry level from interfering with the estimation results. We found our results to be still robust.

Another concern with our identification strategy is that we cluster our sample at the city level, and some cities might only have observations belonging to either the treatment or the control group. This might make companies in certain cities incomparable with other observations. We excluded all observations located in these cities, obtaining a sample size of 38,589, and conducted DID regression once more. We found no significant difference from the baseline regression, proving the robustness of our results. Furthermore, the order of the CIEP might potentially lead to policy shocks impacting all treatment groups at varying times, severely interfering with the estimation results of the DID model. We introduced a dummy variable representing whether a province was inspected by the CIEP in 2016 and employed a triple difference model. We found that the order of the CIEP did not have a significant impact, eliminating potential estimation biases.

## 3.3 Variables

**3.3.1 Dependent variable: CP.** If the annual report disclosed by the listed company reveals public welfare or non-public welfare donations, the sum of the two is recorded as *CP*

after taking the logarithm. If there are no charitable donations or disclosures, then CP = 0. The presence of donations (recorded as *CP_Dummy*) and the ratio of donations to assets (= Total Donations / Total Assets × 100%, recorded as *CP_Ratio*) are used as alternative measures in the robustness tests.

**3.3.2 Explanatory variable: GEA.** Previously, text analysis based on public policy documents was primarily adopted to measure GEA. This included using word frequency statistics to illustrate the distribution of the government's attention to environmental concerns at varied times [96, 97]. Shi et al. [98] collected over 600 reports of provincial governments from 1998 to 2017, statistically analyzing the phrases related to environmental protection in these reports, thereby examining how GEA was distributed. Due to the reliance on the statistician's subjective judgment in terms of the selection of words, the method of word frequency statistics lacks credibility. Local governments may outwardly express compliance but may not genuinely prioritize their attention, leading to a significant bias in measures. Furthermore, the "planning effect" could occur, where specific phrases are excessively mentioned in the initial year of each five-year plan, causing a bias owing to the information intensity [99]. This study opts to use an exogenous shock as a quasi-natural experiment to measure GEA, which can overcome the subjectivity of text analysis.

According to the "Industry Classification Guide for Listed Companies", revised by the China Securities Regulatory Commission in 2012, the "Industry Classification Management Directory for Environmental Protection Inspection of Listed Companies", formulated by the Ministry of Environmental Protection in 2008, and the "Guidelines for Environmental Information Disclosure by Listed Companies", formulated by the Ministry of Environmental Protection in 2010, this study defines 16 industries as heavily polluting industries, including thermal power, steel, cement, aluminum electrolysis, coal, metallurgy, the chemical industry, petrochemicals, building materials, papermaking, brewing, pharmaceuticals, fermentation, textiles, leather making, and mining. Treat = 1 if the company belongs to a heavily polluting industry; otherwise, Treat = 0. Time = 1 if the time is after the implementation of the CIEP (i.e., 2016 and onwards); otherwise, Time = 0. As an explanatory variable,

GEA = Treat × Time. This study's core concern is $\hat{\beta}_1$, the coefficient of GEA. If $\hat{\beta}_1 > 0$, it indicates that companies in heavily polluting industries have increased their CP after the CIEP, as compared to companies in non-heavily polluting industries. If $\hat{\beta}_1$ is not significantly greater than 0, it indicates that CP do not increase.

**3.3.3 Control variables: X.** Based on the existing literature [52, 100], this study selects Firm Age, Nature of Equity, Number of Independent Directors, Herfindahl Index, Shareholding Ratio of the Largest Shareholder, Return on Assets (ROA), Debt-to-Assets Ratio, Assets Size, and Logarithm of Monetary Cash as control variables, defined as follows:

1. Firm Age: Calculated by subtracting the year of the company's establishment from the current year.

2. Nature of Equity: If the company is an SOE, the value is 1; if it is an NSOE, the value is 0.

3. Independent Directors: The number of independent directors directly extracted from corporate annual financial reports.

4. Herfindahl Index: Calculated according to the following formula,

    $HI = \sum_{t=1}^{n} (X_i/X)^2$, $X = \sum_{t=1}^{n} X_i$, where $X_i$ is the annual operating income of company i in the industry, n is the number of companies in the industry, and the data for both $X_i$ and n are extracted from the annual reports of A-share listed companies.

5. Shareholding Ratio of the Largest Shareholder: Directly extracted from corporate annual financial reports.

6. ROA: = Net Profit / Total Assets × 100%.

7. Debt-to-Assets Ratio: = Total Liabilities / Total Assets × 100%.

8. Assets Size: = Ln (Total Assets + 1)

9. Logarithm of Monetary Cash: = Ln (Monetary Cash + 1)

In addition, Model (1) also controls for industry fixed effects δ, regional fixed effects ω, and year fixed effects μ, and ε is the estimated residual term. Only two control variables, Firm Size and Logarithm of Monetary Cash, are nominal variables in the multiple regression model. Thus, the interference they may cause to the regression results is relatively light. Moreover, Model (1) controls for year fixed effects μ, which can alleviate potential issues to a certain extent. Notably, Treat and Time would each be absorbed by industry fixed effects δ and year fixed effects μ, they are not included separately in Model (1).

**3.3.4 Other variables.** The other variables that will be employed in the discussion of mechanisms and heterogeneity, as well as robustness tests, include the following:

1. Rectifying Cases: Manually collected from the official website of MEEC, calculated after aggregating at the provincial level.

2. Backward: Manually collected from the official website of MEEC. If the inspection team conducts a follow-up inspection on a province, the value is 1; otherwise, the value is 0.

3. ESG Performance: The China Securities ESG rating is categorized into 9 levels and is rated four times a year. This study assigns a value of 1–9 to measure the ESG performance of the company for that year, with the average score being used as the indicator.

4. Firm Size: According to the classification standard set by the National Bureau of Statistics, sample companies with a total number of employees equal to or greater than 2000 are defined as large companies, represented by a value of 1. Those that do not meet this criterion are defined as small and medium-sized companies, represented by a value of 0.

5. SO2 Emissions: Data are sourced from the China Statistical Yearbook and China Environmental Statistics Yearbook. Statistics are conducted at the provincial level. If the amount is greater than or equal to the median, it is defined as a high-pollution area with a value of 1; if it is less than the median, it is defined as a low-pollution area with a value of 0.

6. Pollution Control Expenditures: Data are sourced from the CNRDS database. Statistics are conducted at the company level. If the amount is greater than or equal to the median, it is defined as a high-pollution cost company with a value of 1; otherwise, it is a low-pollution cost company with a value of 0.

7. Proportion of Long-term Loans in Total Assets: = Long-term Liabilities / Total Assets × 100%.

8. Proportion of Interest Expenditure in Total Liabilities: = Debt Interest Expenditure / Total Liabilities × 100%.

**Table 2. Descriptive statistics.**

| Variable | Mean | SD | Min | Max |
|---|---|---|---|---|
| CP | 8.58 | 6.14 | 0 | 20.65 |
| CP_Dummy | 0.68 | 0.47 | 0 | 1 |
| CP_Ratio | 0.02 | 0.07 | 0 | 3.17 |
| Firm Age | 9.85 | 7.07 | 0 | 32 |
| Nature of Equity | 0.42 | 0.49 | 0 | 1 |
| Independent Directors | 3.18 | 0.63 | 0 | 8 |
| Herfindahl Index | 0.20 | 0.18 | 0.03 | 1 |
| Shareholding Ratio of the Largest Shareholder | 35.64 | 15.42 | 0.29 | 100 |
| ROA | 0.04 | 0.07 | -0.30 | 0.21 |
| Debt-to-Assets Ratio | 0.43 | 0.21 | 0.05 | 0.97 |
| Assets Size | 21.93 | 1.33 | 10.84 | 28.64 |
| Logarithm of Monetary Cash | 20.04 | 1.47 | 7.69 | 26.51 |

# 4. Results

## 4.1 Descriptive statistics

Table 2 displays the descriptive statistics of all the main variables. Notably, 68% of the sample companies have engaged in CP. The average value of CP was 8.58, with a standard deviation of 6.14, indicating that there is a large variance among sample companies for this indicator.

## 4.2 Baseline regression

According to Model (1), OLS regression is used, and Table 3 presents the results. No control variables are included in the model; only DID is performed, as shown in Column 1. The coefficient of GEA is found to be 0.368 and is significantly positive. As shown in Column 2, the regression model includes control variables at the company level, and the coefficient is 0.383, also significantly positive. The model further controls for city fixed effects, industry fixed effects, and year fixed effects, as seen in Column 3. The results show that the coefficient of GEA is 0.431 and remains significantly positive, indicating that GEA indeed enhances CP. This result holds even after considering non-time-varying differences in corporate characteristics, industries, and regions, as well as inherent differences between years. This confirms Hypothesis 1. However, the underlying mechanisms still require further discussion in the following sections.

## 4.3 Hypothesis tests

**4.3.1 Parallel trend test.** A critical aspect of the DID model is the parallel trend hypothesis, which states that before the implementation of the CIEP, heavily polluting companies (as

**Table 3. Baseline regression.**

| Dependent Variable: CP | (1) | (2) | (3) |
|---|---|---|---|
| GEA | 0.464*** | 0.456*** | 0.507*** |
| | (2.76) | (2.91) | (3.25) |
| Control Variables | No | Yes | Yes |
| Fixed Effects (City, Industry, Year) | No | No | Yes |
| Observations | 40,406 | 40,406 | 40,406 |

the treatment group) and non-heavily polluting companies (as the control group) exhibit a consistent time trend. This study tests whether the parallel trend hypothesis is met to ensure the reliability and stability of the estimation results. Following Jacobson et al. [101], we employed the event study method to conduct a parallel trend test, estimating as follows:

$$CP_{ijct} = \alpha + \sum_{k=-6}^{k=-1} \beta_k D_{ijc,t_0+k} + \gamma D_{ijc,t\geq0} + X'_{ijct}\varphi + \delta_j + \omega_c + \mu_t + \varepsilon_{ijct} \qquad (2)$$

$$CP_{ijct} = \alpha + \sum_{k=-6}^{k=4} \beta_k D_{ijc,t_0+k} + X'_{ijct}\varphi + \delta_j + \omega_c + \mu_t + \varepsilon_{ijct} \qquad (3)$$

D is a series of dummy variables, representing the year k after heavily polluting companies were subjected to the CIEP. k = 0 when it occurs in 2016. $\beta_k$ represents the difference in CP between the treatment and control groups in year k after the CIEP was introduced. If the trend of $\beta_k$ during the period of k < 0 is relatively flat, this proves that it meets the parallel trend hypothesis. Otherwise, it proves that the treatment and control groups have already experienced significant differences prior to the policy's implementation and do not meet the parallel trend hypothesis. According to Models (2) and (3), the OLS regression is performed, and other control variables are consistent with the baseline regression. To avoid the collinearity problem, we take the period of k = −1 as the baseline period. Table 4 presents the results.

**Table 4. Event study method.**

| Dependent Variable: CP | (1) | (2) |
|---|---|---|
| D(t≥0) | 0.701*** | |
| | (4.76) | |
| D(t = -6) | 0.159 | 0.160 |
| | (0.66) | (0.66) |
| D(t = -5) | 0.207 | 0.208 |
| | (0.92) | (0.92) |
| D(t = -4) | 0.311 | 0.312 |
| | (1.16) | (1.16) |
| D(t = -3) | 0.599*** | 0.600*** |
| | (2.87) | (2.88) |
| D(t = -2) | 0.149 | 0.150 |
| | (0.68) | (0.68) |
| D(t = 0) | | 0.718*** |
| | | (3.64) |
| D(t = 1) | | 0.622*** |
| | | (3.19) |
| D(t = 2) | | 0.878*** |
| | | (4.49) |
| D(t = 3) | | 0.602*** |
| | | (3.26) |
| D(t = 4) | | 0.696*** |
| | | (3.89) |
| Control Variables | Yes | Yes |
| Fixed Effects (City, Industry, Year) | Yes | Yes |
| Observations | 40406 | 40406 |

As shown in Column 1 of Table 4, the coefficients of each period before the implementation of the CIEP are not significant, indicating that prior to the implementation of the CIEP (t < 0), the difference in CP between the treatment and control groups is essentially similar to that in the baseline period. This supports the parallel trend hypothesis. The only exception is the period t = −3, that is 2013. This might be attributed to the change in China's leadership in 2013, and the comprehensive change in politics accidentally increased the policy's uncertainty. Thus, the tendency to increase CP emerged in this period alone. However, the coefficients of D (t ≥ 0) are still significant at the 1% level, consistent with the previous reslut.

As shown in Column 2 of Table 4, based on Model (3), we find that the coefficients of each period of D (t < 0) are still not significant (t = −3, i.e.,2013, is still an exception), whereas the coefficients of each period of D (t ≥ 0) are significant at the 5% or 1% level. This further indicates that, before the implementation of the CIEP, the difference in CP between the treatment and control groups was comparable to the baseline period. However, in the years after the CIEP was introduced, heavily polluting companies significantly increased CP.

After extracting the annual estimated coefficients from the regression results of Model (3), they are plotted with a 95% confidence interval, as shown in Fig 2. Prior to the implementation of the CIEP, we observe no significant difference in trends between the treatment and the control groups, and the 95% confidence interval always includes 0 (except for 2013). However, starting from 2016 and onwards, the 95% confidence interval does not include 0 and is significantly greater than 0. This indicates that GEA has enhanced CP and the parallel trend hypothesis is satisfied.

**4.3.2 Placebo test.** Another issue in the DID model is the presence of unobservable corporate characteristic factors that change over time. This omission can lead to estimation bias. Different companies have many distinct characteristics. Although our identification strategy includes the fixed effects of the city in which the company is located and industry fixed effects to control for the impact of unobservable characteristics that remain constant over time on CP, these characteristics may also produce different impacts as time passes, thereby interfering with the identification hypotheses. These impacts cannot be directly controlled by the model in this study. In response to this, following Chetty et al. [102], this study adopts an indirect placebo test. First, a fictitious explanatory variable GEA$^{fake}$ is randomly assigned to each observation, creating a fictitious DID regression and a fictitious estimation coefficient $\hat{\beta}^{fake}$ is

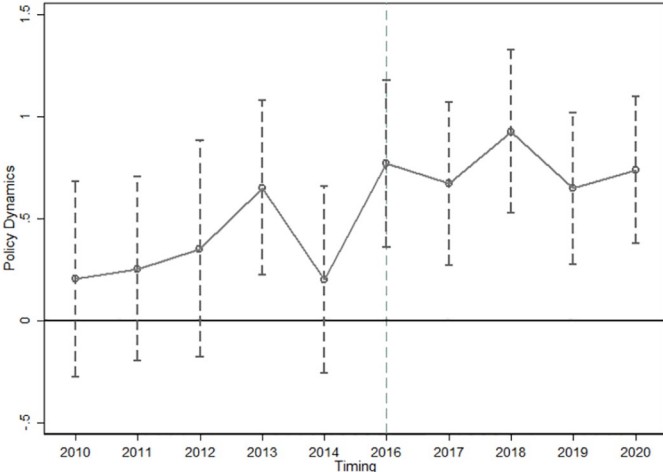

**Fig 2. 95% confidence interval of policy effect.**

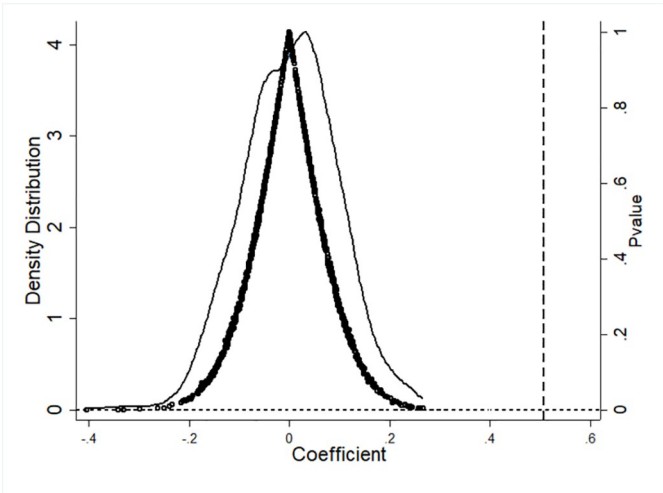

**Fig 3. Placebo test.**

calculated. This process is repeated 1000 times to obtain 1000 fictitious coefficients. If the randomly generated $\hat{\beta}^{fake}$ indeed affects the estimation results, that is, the mean value of $\hat{\beta}^{fake}$ is significantly different from 0, it proves that the estimation is biased and suggests that there indeed exist omitted unobservable factors. Fig 3 presents the distribution of $\hat{\beta}^{fake}$, and we find that the mean value of $\hat{\beta}^{fake}$ is essentially zero, distributed near zero, and follows a normal distribution. This indicates that the empirical model passes the placebo test, and there are no omitted unobservable factors.

**4.3.3 Propensity score matching-difference in differences.** Another concern about the identification strategy is the comparability between heavily polluting and non-heavily polluting industries. While we control for many corporate characteristic variables and fixed effects, there may still be unknown or unobservable characteristics that render the treatment and control groups incomparable. If there exist some heavily polluting companies with weak propensities for CP that voluntarily withdraw from the heavily polluting industries after the CIEP is introduced, or if there are other self-selections that are not addressed in this study but may potentially exist, then these can lead to an estimation bias. This study adopts the PSM-DID to eliminate this problem and ensure sufficient comparability between the treatment and the control groups.

First, all control variables of the baseline regression are selected as matching covariates. A Probit regression is performed using Treat as the dependent variable, and each regression coefficient of the matching covariate is used as a weight to fit the propensity score value of each observation. This score reflects the probability of an observation being in the treatment group. Then, based on the propensity score values, one-to-one nearest neighbor matching with replacement is performed between the treatment group and the control group, eventually obtaining a control group that matches the treatment group. The PSM in this study meets the balance hypothesis. The propensity score values of the treatment and control groups before and after matching are shown in Figs 4 and 5. In Figs 4 and 5, the horizontal axis represents the propensity score value, and the vertical axis represents the kernel density, which is the data distribution density derived from the data themselves. Compared to the case before matching, we find that the propensity score values of the treatment and control groups are more similar after matching. This implies that the treatment and control groups are comparable in other

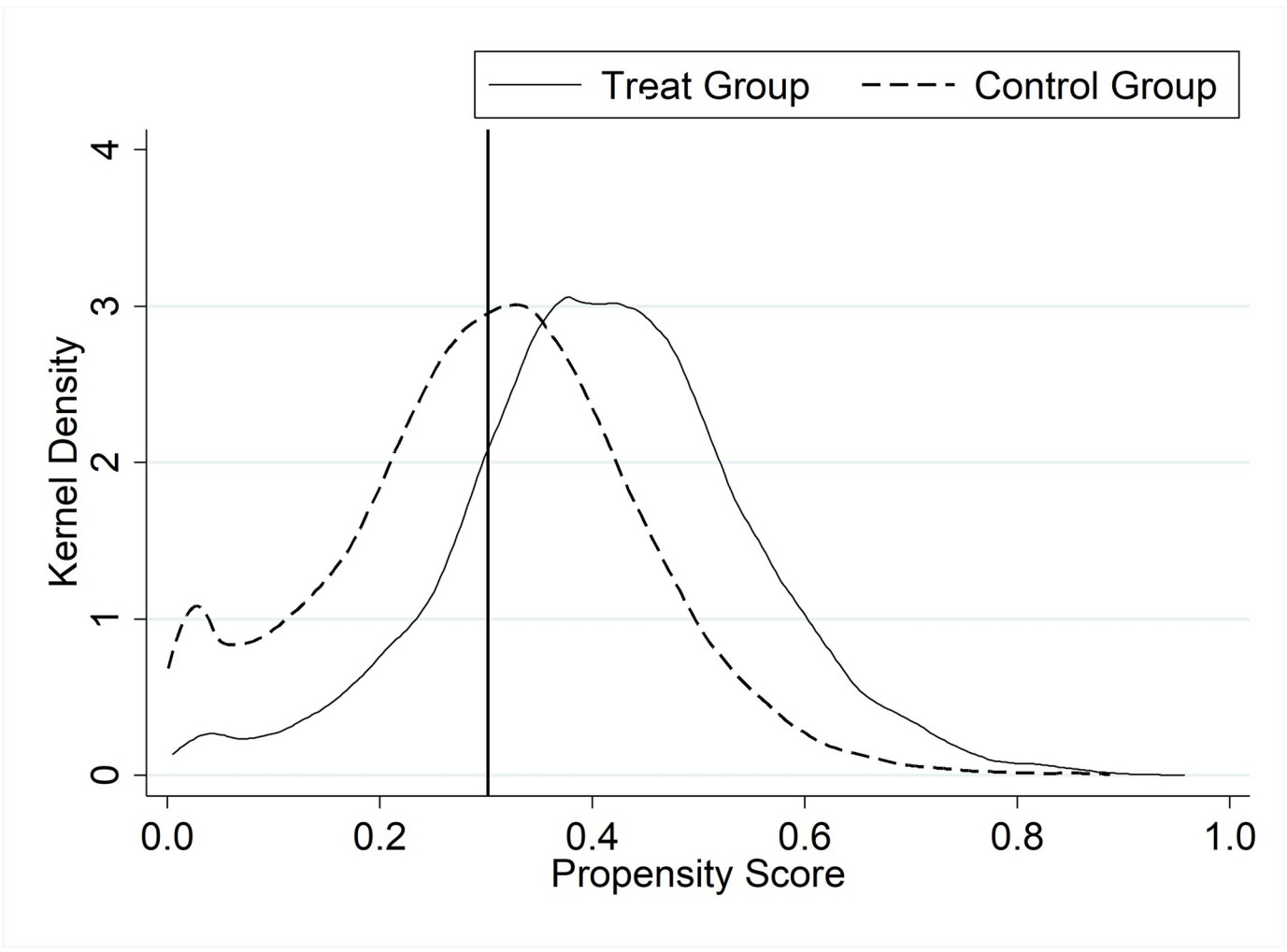

**Fig 4. Propensity score values of treatment and control groups before matching.**

dimensions, and the only significant difference between them is whether they are impacted by the CIEP, which, to some extent, guarantees the reliability of the DID results.

Table 5 presents the comparison of the regression results of the treatment and control groups before and after PSM. Following Shi et al. [103], we transform the panel data into cross-sectional data year by year for matching. Column 1 of Table 5 displays the results of the mixed OLS regression of the cross-sectional data year by year; Column 2 shows the regression results obtained using high-dimensional fixed effects; Column 3 presents the regression results of the sample with non-empty weights after PSM; Column 4 shows the regression results of the sample that meets the common support hypothesis after PSM; and Column 5 shows the frequency-weighted regression results considering the importance of the sample after PSM. We find that GEA is always significantly positive at the 1% level, indicating that, after PSM, the CP of heavily polluting companies still significantly increase after the implementation of the CIEP, and the result of the baseline regression is robust. In addition, we directly match the panel data period by period, following Heyman et al. [104]. The results remain consistent.

**4.3.4 Control for concurrent policies.** *(A) The "de-capacity" policy.* Concurrent with the period leading up to and following the introduction of the CIEP, several other environmental

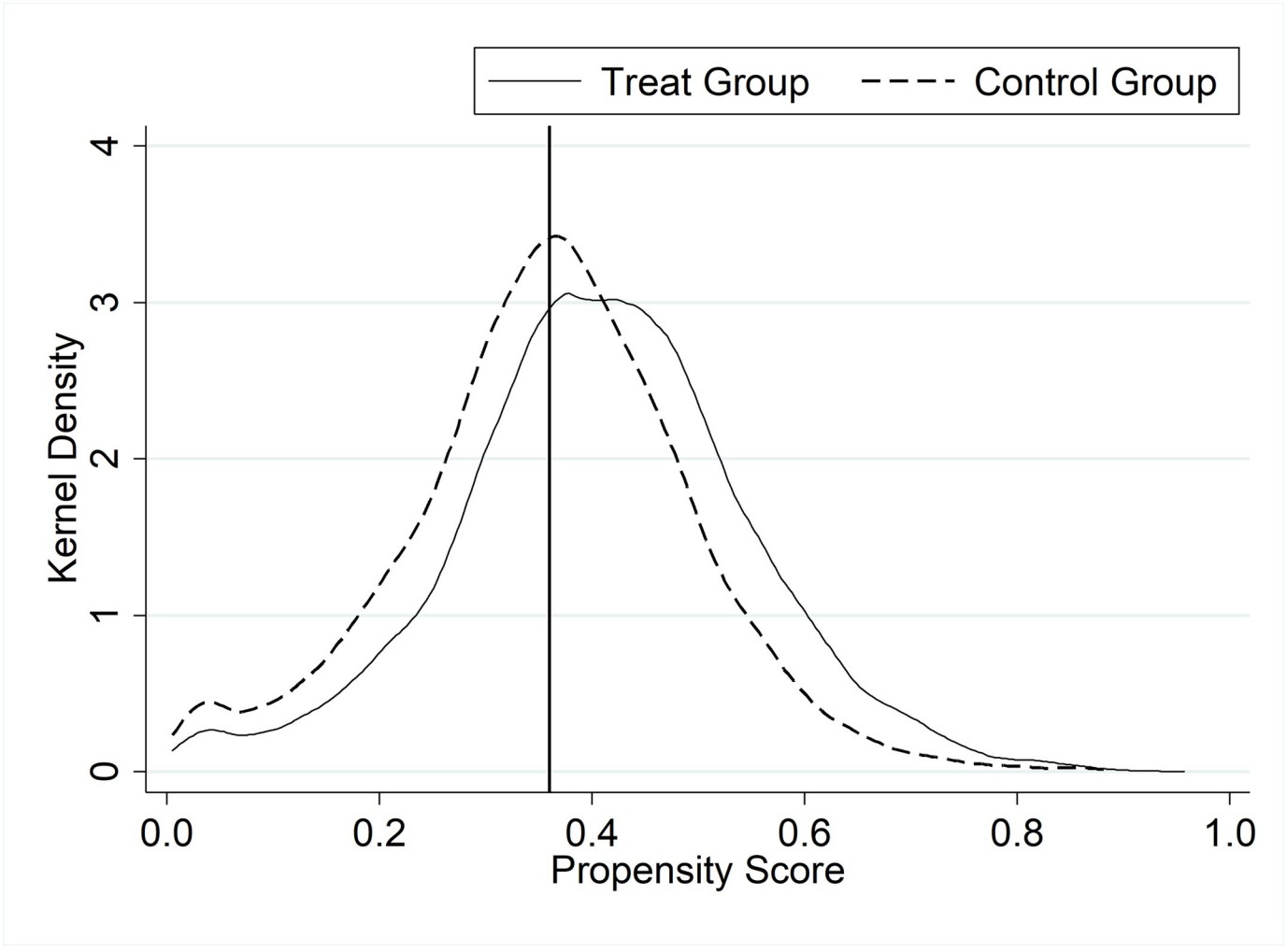

**Fig 5. Propensity score values of treatment and control groups after matching.**

policies may have been implemented, which could potentially influence outcomes related to CP. Consequently, it is imperative to isolate the effects of the CIEP from those of these competing policies. For instance, in October 2013, the State Council of China promulgated the "Guiding Opinions on Resolving the Serious Overcapacity Conflict", mandating a reduction in production capacity within certain industries suffering from overcapacity over a three-to-five-year timeframe. These industries include the steel, cement, electrolytic aluminum, flat glass,

**Table 5. PSM-DID.**

| Dependent Variable: CP | (1) | (2) | (3) | (4) | (5) |
|---|---|---|---|---|---|
| GEA | 0.768*** | 0.768*** | 0.823*** | 0.763*** | 0.824*** |
| | (11.45) | (11.25) | (10.01) | (11.18) | (11.71) |
| Control Variables | Yes | Yes | Yes | Yes | Yes |
| Fixed Effects (City, Industry, Year) | Yes | Yes | Yes | Yes | Yes |
| Observations | 40,406 | 40,406 | 25,047 | 40,358 | 38,817 |

**Table 6. Robustness tests.**

| Panel A | (1) | (2) | (3) | (4) |
|---|---|---|---|---|
| | CP | CP | CP | CP |
| GEA | 0.394** | 0.503*** | 0.465*** | 0.700*** |
| | (2.44) | (3.23) | (2.80) | (3.96) |
| Control Variables | Yes | Yes | Yes | Yes |
| Fixed Effects (City, Industry, Year) | Yes | Yes | Yes | Yes |
| Observations | 38,423 | 40,406 | 40,406 | 40,406 |
| Panel B | (1) | (2) | (3) | (4) |
| | CP | CP_Dummy | CP_Ratio | CP |
| GEA | 0.525*** | 0.173*** | 0.005** | 0.503** |
| | (3.27) | (2.96) | (2.22) | (2.18) |
| GEA×Done_2016 | | | | 0.069 |
| | | | | (0.22) |
| Control Variables | Yes | Yes | Yes | Yes |
| Fixed Effects (City, Industry, Year) | Yes | Yes | Yes | Yes |
| Observations | 38,589 | 40,406 | 40,406 | 40,406 |

and shipbuilding sectors that significantly intersect with those identified as heavy polluters. The interplay between these sectors may result in an overestimation of the impact of GEA.

To mitigate potential contamination of the "de-capacity" policy's influence from 2013 to 2018, this study excludes observations of companies operating within the steel, cement, electrolytic aluminum, flat glass, and shipbuilding sectors from the analysis. Additionally, we incorporate control variables along with city, industry, and year fixed effects to further refine our estimations. As shown in Column 1 of Panel A in Table 6, the result of the OLS regression affirm a significantly positive coefficient of GEA, suggesting that our findings are robust to the exclusion of these potentially confounding factors.

*(B) The green credit policy.* In 2013, the China Banking Regulatory Commission introduced the "Green Credit Statistical System", followed by the "Key Evaluation Indicators for the Implementation of Green Credit" in 2014. These guidelines delineate the parameters of green credit and necessitate the collection of data on loans directed towards firms with significant environmental risks or those engaged in environmental protection projects. The green credit policy has the potential to exacerbate the credit constraints faced by companies with high pollution levels, which could, in turn, influence their propensity towards CP. Following Su and Lian [105], this study incorporates two proxy variables for the green credit policy: the ratio of long-term loans to total assets and the ratio of interest expenditure to total liabilities. The results of the OLS regression, presented in Column 2 of Panel A in Table 6, indicate that even after controlling for the effects of the green credit policy, the promotional impact of GEA on CP remains statistically significant.

**4.3.5 Other robustness tests.** *(A) Incorporating firm fixed effects.* The fixed effects have included city, industry, and year fixed effects. To enhance the robustness of the estimation results, we we propose to further control for firm fixed effects. This addition will account for unobserved firm-level characteristics that remain constant across region, industry, and over time. All other control variables remain consistent with those utilized in the baseline regression. The result of the OLS regression, presented in Column 3 of Panel A in Table 6, reveal that the coefficient of GEA maintains its significantly positive value.

*(B) Accounting for interaction of fixed effects.* To address the factors that vary over time within industries and regions, this study also includes the interaction of industry fixed effects

with year fixed effects and region fixed effects with year fixed effects. The result of this extended model is displayed in Column 4 of Panel A in Table 6. Despite these additional controls, our findings remain robust.

*(C) Utilizing regions with both treatment and control groups.* A further consideration is the clustering of samples at the city level, which may lead to incomparability between companies in cities with observations from only the treatment or control group. To address this, we exclude all observations from such cities, resulting in a refined sample comprising 38,589 observations. We then re-conduct the DID regression, and the result is shown in Column 1 of Panel B in Table 6. The coefficient of GEA is significantly positive at the 5% level, with a numerical value nearly identical to that observed in the baseline regression, reinforcing the validity of our initial results.

*(D) Changing measurement approaches.* To mitigate potential endogeneity issues arising from measurement bias in CP, we employ alternative measures for the dependent variable. Specifically, we use the presence of donations (recorded as CP_Dummy) and the ratio of donations to total assets (= Total Donations / Total Assets × 100%, recorded as CP_Ratio) as surrogate indicators for CP. The results of the OLS regressions using these new measures are presented in Columns 2 and 3 of Panel B in Table 6, consistent with the baseline regression.

*(E) Implementing a triple-difference model.* Given that the sequence of inspection may result in policy shock not affecting treatment groups simultaneously, a standard DID model might produce biased estimates. To address this, we introduce a dummy variable, Done_2016, which indicates whether a province was subject to inspection by the CIEP in 2016. If the province was inspected, Done_2016 = 1; otherwise, Done_2016 = 0. Done_2016 is interacted with GEA in the triple-difference model, while all other control variables remain consistent with the baseline regression. The result is shown in Column 4 of Panel B in Table 6. We find that the coefficient of GEA is still significantly positive, whereas the coefficient of the interaction term, GEA × Done_2016, is not significant. This suggests that the deterrent effect of the CIEP is uniformly distributed across the nation, and there is no unobservable estimation bias resulting from variations in the inspection sequence.

## 5. Discussion

### 5.1 Mechanism

The previous sections have provided reasonable causal evidence of the influence of GEA on CP. This section discusses the mechanism. Based on institutional theory, we posit that the following mechanisms may exist: GEA, as a form of campaign-style institutional change, alters corporate expectations for the future institutional environment, thereby encouraging companies to get more engaged in CP.

One crucial factor in expectation change is whether the institutional uncertainty is severe and persistent. The details of the CIEP collected from the official website of the MEEC encompass these two aspects. First, the inspection team has compiled the number of rectification cases by companies after the implementation of the CIEP in each province, which directly reflects the intensity of the uncertainty caused by GEA. The more rectification cases there are, the higher the intensity of uncertainty, and the harder it is for companies to have a stable expectation for the future. Second, it is crucial to consider if the inspection team adopts a "look-back" measure. The look-back measure is a follow-up inspection carried out by the inspection team since 2018 to consolidate the inspection effect further. Following the first round of inspections in 20 provinces, the inspection team "looked back" at the rectification results, conducting a special inspection on the completion of rectification tasks by local governments. The "look-back" period typically lasts for one month, specifically targeting problems

such as superficial, perfunctory action, temporary shutdowns, and false rectifications that took place during the first round of inspections. The "look-back" as a follow-up insepction reflects the persistence of institutional uncertainty. After the inspection team withdraws, GEA may conduct random checks, and companies cannot predict whether the short-term strategy will be optimal in the long run. We introduce two variables—the number of corporate rectification cases and whether the province was "looked back" at—and multiply them with GEA, respectively, while keeping other control variables the same as in the baseline regression. If this mechanism is in place, we expect the coefficient of the interaction term to be significantly positive. As can be seen from Columns 1 and 2 of Table 7, the more intense and persistent the GEA, the more it will further enhance CP, validating the mechanism of expectation change.

Alternatively, expectation change also relies on the sensitivity of companies to uncertainty. Companies that exhibit a higher sensitivity to uncertainty tend to exert more effort to mitigate potential negative impacts. The endowment effect theory in behavioral economics posits that people assign a higher value to what they own and thus feel greater pain when they lose it [92]. From this standpoint, companies with better legitimacy endowments should be more sensitive. The reasons are as follows: First, companies with better legitimacy endowments usually have larger investments in existing businesses, assets, and infrastructure, which form huge sunk costs. Second, companies with better legitimacy endowments often have a stable market position and high brand value. They are the core endowments of large companies, who are reluctant to see these endowments damaged by uncertainty and need to consider how to meet the expectations of shareholders and stakeholders. Consequently, large companies will be more cautious and conservative and will not ignore uncertainty. Third, such companies often have more complex organizational structures and longer decision-making processes, resulting in delayed responses when facing uncertainty. The endowment effect here is reflected in large companies' reliance on existing organizational structures and workflows. They exhibit a strong resistance to change, as it entails relinquishing the current ways of operation. Furthermore, they often receive more regulation and public attention, resulting in any mistakes or losses in the face of uncertainty being magnified and subjected to more criticism. Their primary focus will be on safeguarding their reputation and public image.

Therefore, companies with better legitimacy endowments are more sensitive to uncertainty, fear potential losses due to lack of legitimacy, and are likely to respond more actively. The most typical examples are SOEs and companies with better ESG performance. These two types of companies clearly possess better legitimacy endowments in the Chinese context. We introduce a dummy variable reflecting whether it is an SOE and the ESG score and multiply them

**Table 7. Mechanism.**

| Dependent Variable: CP | (1) | (2) | (3) | (4) |
|---|---|---|---|---|
| GEA × Rectification_Cases | 0.007*** | | | |
| | (3.05) | | | |
| GEA × "Look-Back" | | 0.571*** | | |
| | | (2.89) | | |
| GEA × SOE | | | 0.647** | |
| | | | (2.37) | |
| GEA × ESG | | | | 0.306*** |
| | | | | (2.90) |
| Control Variables | Yes | Yes | Yes | Yes |
| Fixed Effects (City, Industry, Year) | Yes | Yes | Yes | Yes |
| Observations | 40,406 | 40,406 | 40,406 | 5,366 |

with GEA, respectively. The other control variables are the same as those used in the baseline regression. If the mechanism is valid, we expect the coefficient of the interaction term to be significantly positive. As shown in Columns 3 and 4 of Table 7, GEA significantly enhances CP for SOE and high-ESG-scored companies. It indicates that companies that are more sensitive to GEA are more likely to change expectations, which, to some extent, once again validates the establishment of the mechanism.

## 5.2 Heterogeneity analysis

**5.2.1 Firm size.** The fulfillment of social responsibility by firms of varying sizes may differ in response to GEA, as their needs for legitimacy also differ. Large enterprises, given their significant contributions through local taxes, employment generation, and industrial clustering, play a pivotal role and often face greater demands for legitimacy. They typically consider the long-term strategic direction of the companies and benefit from less stringent financial constraints, enabling them to secure external financing more readily from banks and other financial institutions or through the issuance of stocks and corporate bonds. This financial robustness facilitates engagement in CP. In contrast, small and medium-sized companies face more pronounced financial constraints and, despite potential aspirations to enhance their legitimacy, may lack the necessary resources for effective implementation.

In this study, we adopt the classification standard of the National Bureau of Statistics of China, designating firms with a total employees count of 2000 or more as large companies, while those falling short of this criterion are classified as small and medium-sized companies. We conduct grouped OLS regression, and the results are shown in Columns 1 and 2 of Panel A in Table 8. The results indicate that the promotion impact of GEA on CP is significant only among large companies. This suggests that GEA introduces institutional uncertainty that companies must navigate with financial backing. Large companies, with their more relaxed financial constraints, are better positioned to engage in CP.

**5.2.2 Industry competition.** The response of companies to environmental governance pressure can vary significantly depending on the level of competition within their industry. In highly competitive sectors, companies may increase their investments in CP to mitigate risks, such as reputation damage, increased credit cost, and loss of market share, thereby maintaining or enhancing their corporate legitimacy. Conversely, in monopolistic industries, companies possess considerable pricing power and are less vulnerable to external pressures. They

**Table 8. Heterogeneity.**

| Panel A | (1) | (2) | (3) | (4) |
|---|---|---|---|---|
| Dependent Variable: CP | Large sized | Small and medium-sized | High competition | Low competition |
| GEA | 0.536** | 0.346 | 0.348 | 0.620*** |
| | (2.39) | (1.55) | (1.47) | (3.00) |
| Control Variables | Yes | Yes | Yes | Yes |
| Fixed Effects (City, Industry, Year) | Yes | Yes | Yes | Yes |
| Observations | 19,476 | 20,930 | 19,976 | 20,430 |
| Panel B | (1) | (2) | (3) | (4) |
| Dependent Variable: CP | High-pollution area | Low-pollution area | High environmental expenditure | Low environmental expenditure |
| GEA | 0.497*** | 0.124 | 0.409** | 0.013 |
| | (2.62) | (0.34) | (2.44) | (0.02) |
| Control Variables | Yes | Yes | Yes | Yes |
| Fixed Effects (City, Industry, Year) | Yes | Yes | Yes | Yes |
| Observations | 25,081 | 15,325 | 38,058 | 2,348 |

may be able to absorb environmental pollution penalties without significant impacts on their market share or operational profits, potentially diminishing their incentive to engage in CP.

To examine this, we utilize the Herfindahl index to measure industry competition and categorize our samples into high-competition and low-competition groups based on the median of index value. The results of the grouped OLS regression are shown in Columns 3 and 4 of Panel A in Table 8. Contrary to our expectations, we find that GEA does not significantly promote CP in high-competition industries. One possible explanation for this unexpected finding is that the the ability to manipulate legitimacy is constrained by financial constraints. Companies in high-competition industries face more market pressure, lower profit margins, and stricter finances constraints, which may hinder their capacity to engage in enhancing legitimacy. Despite being theoretically able to withstand higher penalties for non-compliance, companies in less competition, near-monopolistic environments often enjoy larger profits and superior financial conditions, enabling them to be more effectively engaged in CP.

**5.2.3 Regional pollution and environmental expenditure.** For companies operating in heavy pollution industries, financial constraints are not the only factor influencing corporate decisions; governance pressure also plays a crucial role. This pressure manifests in two primary ways. First, in regions with higher pollution levels, companies may feel a greater urgency to enhance their legitimacy due to competitive pressures and public expectations. Second, the sustained burden of high environmental protection costs can impose significant financial strain, providing further incentive for companies to pursue legitimacy.

To investigate these dynamics, we categorize samples into high-pollution area and low-pollution area groups based on the SO2 emissions of each province as of the end of 2015. The results of the grouped OLS regression are shown in Columns 1 and 2 of Panel B in Table 8. We find that GEA significantly enhances CP in high-pollution areas, whereas its impact is not significant in low-pollution areas. This supports our hupothesis that companies in more polluted regions are more proactive in adopting CP and seeking corporate legitimacy.

Additionally, we divide the samples into high-environmental-expenditure and low-environmental-expenditure groups using the median expenditure as a threshold. The results of the grouped OLS regression are shown in Columns 3 and 4 of Panel B in Table 8. We find that the positive impact of GEA on CP is only significant in high-environmental-expenditure groups, suggesting that companies with greater environmental expenditure, likely due to higher levels of pollution, face more acute cost pressure and thus have a stronger demand for legitimacy to navigate the uncertainties associated with institutional changes.

## 6. Conclusion

Environmental pollution is a significant issue in emerging markets. China, being the largest emerging market in the world, is a classic example. The extensive development model of China over the past few decades has led to severe environmental pollution. With the trend of global environmental protection, the Chinese government has placed an increasing emphasis on environmental governance, gradually transitioning to high-quality development and striving to achieve a more sustainable economic development model. Before the Chinese central government forcefully implemented GEA, local officials, driven by personal ambition, generally perfunctorily carried out environmental governance. Although environmental protection was formerly included as one of the "one-vote veto" performances in official assessments, local officials would simply meet the minimum requirements and would not allocate any resources to environmental governance. After all, the return from economic development is unlimited. Therefore, the Chinese central government implemented the CIEP in 2016. This initiative included deploying high-level inspection teams to conduct random inspections in various

provinces. The government forced companies to rectify, interviewing and holding local officials accountable to ensure compliance. It demonstrates the focused efforts of the Chinese government in the environmental sector. Simultaneously, this campaign-style institutional change also brought great uncertainty to companies.

This study utilizes data from the CSMAR, CNRDS, and Wind databases, as well as details of the CIEP manually collected from the official website of MEEC. This study constructs a DID model, using the implementation of the CIEP as a quasi-natural experiment to measure GEA, and examines the impact of GEA on CP. This study primarily draws three significant conclusions: (1) GEA significantly promotes CP, and the results from a series of tests based on different identification hypotheses and alternative measures show robustness. (2) As a form of campaign-style institutional change, GEA changes corporate expectations for the institutional environment, thereby promoting CP. There are two dimensions. On the one hand, it is the intensity of changes. The more intense and persistent the change, the greater the uncertainty it brings, incentivizing companies to engage more in CP. On the other hand, it is the sensitivity of companies to changes. SOEs and high-ESG-scored companies, which originally had better legitimacy endowments, will actively engage in CP to ensure their legitimacy. (3) There is a huge heterogeneity among companies. For large companies, companies in low-competition industries, companies located in high-pollution areas, and high-pollution-expenditure companies, GEA is more likely to enhance CP.

This study aims to provide theoretical guidance on how the government could guide CP. First, this study identifies and highlights the non-transactional political motivation of CP. The traditional view is that Chinese companies engage in CP primarily to build political connections with local officials for direct benefits, such as preferential policies and market access, even if their motivations are not for economic or managerial purposes. This is referred to as transactional political motivation. However, this study reveals that in certain institutional environments, CP is not merely for direct benefits but to establish and uphold corporate legitimacy, align with prevailing institutional norms in the Chinese context, and manage the uncertainty arising from unpredictable institutional changes. This study's results are significant for understanding the complexity of CP and can help the government determine how to guide CP from the perspective of non-transactional political motivation. Second, this study examines the impact of institutional uncertainty on CP. In an environment characterized by high institutional uncertainty, companies often struggle to predict the trajectory of government policies, thus generating a motivation to mitigate risk. This study reveals that GEA, as a signal, can change corporate expectations of the institutional environment. When the government exhibits high environmental attention, companies will expect that the government may adopt stricter environmental policies. This, in turn, generates a stronger risk aversion motivation and then increases CP inputs. Companies are willing to gain approval from the government and minimize the impact that uncertainty may cause. The revelation of this mechanism offers a new perspective on comprehending how companies make strategic decisions in uncertain environments and aids the government in formulating more effective environmental governance.

Several emerging markets have imperfect institutions. Moreover, the market mechanism for economic development is often hindered by significant government intervention. China, with its substantial economic volume, serves as a highly representative source of empirical research evidence. Based on this study's results, three policy recommendations are proposed. First, the government should use campaign-style governance cautiously to avoid disruptions in the institutional environment. Campaign-style governance often exhibits characteristics of concentration and compulsion. While it may achieve notable outcomes in a certain period, it may ultimately increase the expectations of policy uncertainty in the long run, which may

prompt companies to adopt short-sighted strategic behavior. Hence, it is essential for the government to prioritize the continuity and stability of policies, providing companies with a clear and stable institutional environment through long-term mechanisms. Second, although non-transactional political motivation is unrelated to corruption, CP under this motivation may result in excessive non-productive expenditures. Although these expenditures assist companies in managing uncertainty and maintaining legitimacy, they may deplete resources for productive investments, such as innovation and environmental protection. Therefore, the government should implement measures to guide companies to invest more resources in productive activities. This can be achieved by incentivizing companies to engage in research and development of environmental technologies through tax incentives and financial subsidies. Such measures will result in a win-win situation for environmental protection and corporate development. Finally, the government should strongly advocate the value orientation of actively fulfilling corporate social responsibility and use diversified methods to encourage companies to give back to society. This encompasses the following: increasing public awareness and social status of corporate social responsibility through media exposure and social recognition; establishing and enhancing the evaluation system of corporate social responsibility, making it a pivotal criterion for evaluating corporate performance; and facilitating companies' involvement in social welfare initiatives to bolster their reputation, particularly those pertaining to environmental protection.

## Supporting information

**S1 Data.**
(ZIP)

## Author Contributions

**Conceptualization:** Dongli Cao, Chunxian Nie.

**Data curation:** Chunxian Nie.

**Formal analysis:** Dongli Cao.

**Funding acquisition:** Chunxian Nie.

**Investigation:** Chunxian Nie.

**Methodology:** Dongli Cao.

**Project administration:** Chunxian Nie.

**Resources:** Chunxian Nie.

**Software:** Chunxian Nie.

**Supervision:** Dongli Cao.

**Validation:** Dongli Cao.

**Visualization:** Chunxian Nie.

**Writing – original draft:** Dongli Cao.

**Writing – review & editing:** Dongli Cao.

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
