## [Decision Letter · Decision Letter 0]

24 Mar 2024

PONE-D-24-00721The effect of government’s environmental attention on corporate charitable donations based on the credible commitments mechanism: evidence from China’s heavily polluting enterprisesPLOS ONE

Dear Dr. Cao,

Thank you for submitting your manuscript to PLOS ONE. After careful consideration, we feel that it has merit but does not fully meet PLOS ONE’s publication criteria as it currently stands. Therefore, we invite you to submit a revised version of the manuscript that addresses the points raised during the review process.

After a thorough review of the manuscript titled "The effect of government’s environmental attention on corporate charitable donations based on the credible commitments mechanism: evidence from China’s heavily polluting enterprises," and considering the insightful comments and suggestions for improvement from the reviewers, we have reached a decision regarding its publication. The manuscript presents a novel exploration of how governmental focus on environmental issues influences corporate charitable behavior, using a rigorous methodological approach and presenting findings that could significantly contribute to the literature in corporate social responsibility and environmental economics.

Reviewer 1's Comments:

The innovation of this paper should be more prominently highlighted in the abstract.

A summary of existing research gaps after the literature review would enhance the manuscript, emphasizing this paper's innovative contributions.

A framework diagram in the second section would aid in understanding the paper's structure and arguments.

The paper should delve deeper into the managerial insights/policy implications, comparing its theoretical contributions and application values with existing literature, especially in the conclusion.

Reflect more scientific problems, research status, research methods, and innovations in both structure and content.

A crucial discussion on the differences between this study and others should be included to highlight its academic value and relevance, suggesting specific literature for reference.

A thorough check on the English quality is recommended.

The structure needs significant adjustment, with a suggestion for the author to engage more with relevant literature in journals.

Adherence to the correct journal's guidelines (INTRODUCTION→METHOD→RESULTS→DISCUSSION→CONCLUSION) is necessary for consideration for publication.

Reviewer 2's Comments:

The manuscript needs a more systematic analysis of corporate donation behaviour motives and government influence.

A clarification on the focus on corporate donations over environmental investment is required, presenting a theoretical basis and contributions.

Support for the theoretical analysis appears weak, emphasizing the need for a stronger linkage to existing classical theories.

Formula (1) lacks an introduction to relevant variables, with issues in variable relationships noted.

Some control variables are nominal, posing challenges.

The analysis of theory and mechanism lacks persuasiveness, presenting a significant challenge to the manuscript's impact.

Decision:

Considering the above points, the manuscript requires substantial revisions before it can be considered for publication. The innovative aspects and potential contributions of your study to the field are clear, but the reviewers' comments highlight several areas where improvements are necessary to fully realize the manuscript's value. These include a clearer articulation of the study's innovation in the abstract and after the literature review, improvements in the discussion of managerial insights and policy implications, and more rigorous engagement with the journal's structural guidelines.

In addition, both reviewers emphasize the need for a more robust theoretical grounding and methodological clarity, particularly concerning the chosen focus on corporate donations and the presentation and analysis of data. Addressing these points thoroughly will not only strengthen the manuscript's contributions but also ensure its alignment with the journal's standards and expectations.

Recommendation:

We invite you to revise your manuscript, taking into account all the feedback provided. Please provide a detailed response to each point raised by the reviewers, outlining the changes made to the manuscript accordingly. We believe that addressing these concerns will significantly enhance the manuscript's clarity, depth, and impact, making a valuable contribution to the field.

We look forward to receiving your revised manuscript.

Kind regards,

Maria Alina Caratas, Ph.D

Academic Editor

PLOS ONE

Reviewers' comments:

Reviewer's Responses to Questions

**Comments to the Author**

1. Is the manuscript technically sound, and do the data support the conclusions?

Reviewer #1: Yes

Reviewer #2: Yes

2. Has the statistical analysis been performed appropriately and rigorously? 

Reviewer #1: No

Reviewer #2: Yes

3. Have the authors made all data underlying the findings in their manuscript fully available?

Reviewer #1: Yes

Reviewer #2: Yes

4. Is the manuscript presented in an intelligible fashion and written in standard English?

Reviewer #1: No

Reviewer #2: Yes

5. Review Comments to the Author

Reviewer #1: (1) The innovation of this paper needs to be highlighted in the abstract.

(2) The author should summarize the existing research gaps and highlight the innovation of this paper after completing the literature review.

(3) In the second section, Please provide a framework diagram of this paper

(4) What are the managerial insights/policy implications of this study? Compared with available literature, what are the theoretical contributions and application values of this study? It is suggested to enhance the corresponding discussions in the conclusion part.

(5) Whether in terms of structure or content, this article should reflect more scientific problems, research status, research methods and innovation

(6) The article lacks an important discussion link, in which the author should focus on describing the differences between the article study and other scholars' studies, thus highlighting the relevance and academic value of the article, the following literature should be helpful for your research: (1)Reduction pathways identification of Agricultural Water Pollution in Hubei Province, China. (2) A differential game of water pollution management in the trans-jurisdictional river basin. (3) Coordination of the Industrial-Ecological Economy in the Yangtze River Economic Belt, China.

(7) Please do a thorough check on the English quality of this paper

(8) The structure of the article needs to be greatly adjusted, and it is recommended that the author read more relevant papers in journals

(9) The article was not written following the correct journal's guidelines to be considered for publication. INTORDCUTION→MRTHOD→RESULTS→DISSCUSION→CONCLUSION

Reviewer #2: This manuscript examines the impact of environmental inspections on corporate donation behavior. The empirical evidence of this study is relatively standardized, but the flaw lies in its theoretical rationality.

Some specific suggestions are as follows:

Firstly, the effort made in the literature for this manuscript is insufficient. This requires a systematic analysis of the motives behind corporate donation behavior and the influence of government factors. Furthermore, there is no updated literature for this manuscript.

Secondly, why is the research focused on corporate donations rather than corporate environmental investment. This manuscript needs to present its theoretical basis and contributions from this perspective.

Thirdly, theoretical analysis lacks support. This is mainly the author's viewpoint, rather than an extension of existing classical theories. In addition, this suggestion provides a framework diagram of the relationship between classical theories, theoretical extensions, and research hypotheses.

Fourthly, formula (1) does not introduce the relevant variables. In addition, there is a complete multiple contribution between dummy variables and fixed effects, and some dummy variables are not needed.

Fifthly, some variables in the control variables are nominal variables. This may present challenges.

Sixth, the mechanism lacks persuasiveness.

The biggest challenge of this manuscript lies in the analysis of theory and mechanism.

6. PLOS authors have the option to publish the peer review history of their article (what does this mean?). If published, this will include your full peer review and any attached files.

Reviewer #1: No

Reviewer #2: No

---

## [Author Response · Author response to Decision Letter 0]

9 May 2024

Dear Reviewers,

I hope this letter finds you well. I am writing to express my gratitude for the thorough reviews of my manuscript. I am grateful for the time and effort reviewers have dedicated to providing valuable feedback that has significantly enhanced the quality of my research.

I have carefully reviewed the comments and suggestions from two reviewers, and have made the necessary revisions to address each point. I believe that these changes have strengthened the paper's theoretical framework, methodological approach, and empirical findings. Below is a summary of the revisions I have made.

To Reviewer-1:

(1)We have emphasized the innovation of this paper in the abstract. Please see Para 1 at Page No.2

(2)From the perspective of political motivations of corporate philanthropy in the Chinese context, we have revisited the literature review and extensively rewritten the introduction. We have analyzed the shortcomings of relevant literature, summarized the gaps in existing research, and highlighted the innovation of this paper after presenting the main conclusions. Please see Para2 at Page NO.9 and Page 12-14.

(3)We have supplemented a framework diagram of this paper. Please see Page 22

(4)We have expanded the discussion of the theoretical contributions and policy implications of this study in the conclusion part. Please see Page 51-53.

(5)This paper has undergone almost comprehensive revisions to the introduction, theoretical analysis, method, discussion of mechanism, and the conclusion, with adjustments made to the structure. In particular, the introduction has been rewritten following a logical flow of background, research problems, current research status, research methods, main conclusions, and innovations. The other sections have also undergone significant modifications. Now this paper could reflect more scientific problems, research status, research methods, and innovations.

(6)Following the style of several recommended literature, we have added a comparative analysis of differences between this study and other scholars' studies, focusing on the unique theoretical contributions of this paper. Please see Page 12-14.

(7)We have thoroughly reviewed and improved the English quality of this paper.

(8)After reading many relevant papers in journals, we have made substantial adjustments to the structure of this paper.

(9)Following the correct journal’s guidelines, we have adjusted the structure of this paper in the order of "Introduction (and Theory Analysis) → Method→ Results→ Discussion→ Conclusion".

To reviewer-2:

(1)From the perspective of non-transactional political motivations of corporate philanthropy and the impact of campaign-style institutional uncertainty, we have updated numerous latest references, reorganized the literature review, and conducted a systematic theoretical analysis. Please see the rewritten introduction at Page 3-14.

(2)In the Chinese context, the government plays a dominant role in shaping corporate behaviors and granting legitimacy. Corporate philanthropy, a non-productive corporate behavior, is the most important strategic tool for Chinese companies to manipulate legitimacy and respond to institutional uncertainty. Therefore, this paper focuses on corporate philanthropy rather than corporate environmental investment, which is with productive purposes. From this perspective of institutional theory, this paper proposes a framework of theoretical analysis. Please see Para 1-2 at Page No.6 and Page 12-14.

(3)We have rewritten the theoretical analysis part, analyzing the institutional environment and the problems faced by Chinese companies, extending the classical theory of motivations of corporate philanthropy. We have also added a framework diagram to facilitate understanding of the relationship between the theoretical basis, theoretical extensions, and research hypotheses in this paper. Please see Page 17-22.

(4)We have added the introduction to relevant variables in Model (1). Since some unnecessary dummy variables will be absorbed by industry and year fixed effects, we have removed these dummy variables from Model (1). Please see Para 3 at Page 24, Para 1 at Page 25, and Page 27-29

(5)Although firm size and logarithm of monetary cash are nominal variables in the control variables, there are only two nominal variables in the multiple regression model, which may cause minor interference. Furthermore, Model (1) controls for year fixed effects, which we believe could alleviate potential issues to some extent.

(6)We have re-discussed the mechanism and proposed that the mechanism should be expectation changing, which has been validated from the perspectives of the intensity of changes and the sensitivity of companies to the change. We believe this mechanism is reasonable, consistent with the revised theoretical framework, and has persuasiveness. Please see Page 43-45.

I have also made several other minor adjustments to improve the clarity and readability of the paper. I have included the revised version of the paper, labeled as "Revised Manuscript with Track Changes " for your review.

Thank you once again for your valuable comments and suggestions. I hope that the revised paper will meet the standards and expectations of the journal and be considered for publication.

Best regards,

Dongli Cao

---

## [Decision Letter · Decision Letter 1]

15 Aug 2024

Effect of government’s environmental attention on corporate philanthropy based on the institutional theory: Evidence from China’s heavily polluting companies

PONE-D-24-00721R1

Dear Dr. Cao,

We’re pleased to inform you that your manuscript has been judged scientifically suitable for publication and will be formally accepted for publication once it meets all outstanding technical requirements.

Kind regards,

Wei Liu

Academic Editor

PLOS ONE

Additional Editor Comments (optional):

I think this submission can be accepted.

Reviewers' comments:

Reviewer's Responses to Questions

**Comments to the Author**

1. If the authors have adequately addressed your comments raised in a previous round of review and you feel that this manuscript is now acceptable for publication, you may indicate that here to bypass the “Comments to the Author” section, enter your conflict of interest statement in the “Confidential to Editor” section, and submit your "Accept" recommendation.

Reviewer #2: All comments have been addressed

2. Is the manuscript technically sound, and do the data support the conclusions?

Reviewer #2: Yes

3. Has the statistical analysis been performed appropriately and rigorously? 

Reviewer #2: Yes

4. Have the authors made all data underlying the findings in their manuscript fully available?

Reviewer #2: Yes

5. Is the manuscript presented in an intelligible fashion and written in standard English?

Reviewer #2: Yes

6. Review Comments to the Author

Reviewer #2: The author handled or responded to my comment and I have no additional suggestions.

The empirical analysis of this manuscript is reliable, no academic immorality has been found, and I think it can be published.

7. PLOS authors have the option to publish the peer review history of their article (what does this mean?). If published, this will include your full peer review and any attached files.

Reviewer #2: No

---

## [Editor Report · Acceptance letter]

26 Aug 2024

PONE-D-24-00721R1 

PLOS ONE

Dear Dr. Cao, 

I'm pleased to inform you that your manuscript has been deemed suitable for publication in PLOS ONE. Congratulations! Your manuscript is now being handed over to our production team.

Kind regards, 

on behalf of

Prof. Wei Liu 

Academic Editor

PLOS ONE